# The Impact of Oxford Nanopore Technologies Based Methodologies on the Genome Sequencing and Assembly of Romanian Strains of *Drosophila suzukii*

**DOI:** 10.3390/insects16010002

**Published:** 2024-12-24

**Authors:** Attila Cristian Ratiu, Adrian Ionascu, Nicoleta Denisa Constantin

**Affiliations:** 1*Drosophila* Laboratory, Department of Genetics, Faculty of Biology, University of Bucharest, 060101 Bucharest, Romania; attila.ratiu@bio.unibuc.ro (A.C.R.); constantin.nicoleta-denisa@s.bio.unibuc.ro (N.D.C.); 2The Research Institute of the University of Bucharest, 050095 Bucharest, Romania; 3Faculty of Biology, University of Bucharest, Splaiul Independentei 91-95, 050095 Bucharest, Romania

**Keywords:** *Drosophila suzukii*, ONT sequencing, genome assembly, genome scaffolding, natural population

## Abstract

The insect pest *Drosophila suzukii* is considered a growing concern due to agricultural and economic implications, creating an urgent necessity to set out a genotypic foundation with the most modern strategies devised to hinder its negative impact. This study presents the first sequencing project performed in Romania, aiming to characterize two stabilized *D. suzukii* lines derived from natural populations. Additionally, this study addresses insights and strategies in order to maximize the genome assembly quality. An original scaffolding procedure prototype, dScaff, was implemented for genomic graphical representations and to create a basic linear coverage of a genome of similar quality to the current genome assembly.

## 1. Introduction

The species of the genus *Drosophila* are not generally considered pests due to their preference to lay eggs in damaged and rotten fruits. In contrast, *Drosophila suzukii* (Matsumura), a member of the melanogaster species group, oviposit in ripened berries and stone fruits, thus greatly affecting crop productivity and salability. *D. suzukii* was originally reported in Japan, but since the late 2000s it has emerged as a major invasive pest in both Europe and the Americas [1,2]. Worldwide, in agricultural areas *D. suzukii* causes significant yield losses amid soft fruit cultivations, ranging between 25% and 80% of the total production [3], with yearly costs exceeding EUR one billion in the present [4]. *D. suzukii* rapidly colonized Europe’s mainland and its coastal areas, from Spain [1] to Greece [5].

Currently, both the behavior and phenotypic traits of *D. suzukii* are under intense scrutiny within laboratory settings, and are the subject of studies varying from population dynamics and seasonal biology [6] to adaptive traits and their underlying molecular mechanisms. The first genome release of *D. suzukii* was published in 2013 [7], but it contained several errors and suffered from high fragmentation rates, issues addressed and corrected by one of the latest available genome assemblies, which is based on long-read sequencing [8].

Recent genomics projects prompted the identification of several genes and genomic features that could be relevant for deciphering invasion success and sensory evolution genetic mechanisms in *D. suzukii* [4,9,10]. These endeavors clearly stressed that any manipulative genetic technology has to target the specific genotype of a certain pest population.

Long-read sequencing technologies directly target single DNA molecules, can operate at higher speed, and produce longer reads than next-generation sequencing (NGS) [11]. For example, in the case of Oxford Nanopore Technologies (ONT), the MinION device is able to generate several gigabytes (GBs) of raw sequence data in a matter of hours, and the reads may exceed 100 kilobases (kb) in length [12]. ONT operates with relatively reduced sequencing and equipment costs [13,14] and is routinely used to generate reliable assemblies with a coverage greater than 30× [8,15]. Their latest optimizations allow in theory for a median read accuracy of a Phred quality score of at least 20 (Q20) for about 99% of the reads.

*D. suzukii* is expected to harbor very dynamic transposon landscapes since it displays one of the highest transposable elements (TEs) content among *Drosophilidae* [16]. This particularity could partially explain the versatility of this species via the TEs’ functional and potentially adaptive effects on its host [17]. TEs are reported to promote deleterious effects, being selected through genetic drift, influencing genomic recombination, contributing to the emergence of novel cellular functions, acting on telomeres, shaping adaptative features, and being involved in the speciation process [18]. In addition, TEs have been recently considered as potential regulators that could puzzle out the paradox of invasive species, more specifically, how it can be explained that a small population, with trivial starting genetic diversity, can readily adapt to a new environment [19].

During the summer of 2022, we successfully collected *D. suzukii* individuals from two distinct locations within Bucharest, namely, the Botanical Garden of Bucharest (GB) and the National Research and Development Institute for Plant Protection (ICDPP). Starting from those individuals, we stabilized two lines, symbolized GB-ls-coga4 and ICDPP-ams-1, which are expected to loosely meet the criteria of near-isogenic lines that harbor a relatively homogenous genomic background.

Our study focuses mainly on the genome sequencing and assembly of GB-ls-coga4, with an emphasis on how different genome sequencing and basecalling strategies based on ONT and the corresponding data amend the attributes of Canu *de novo* genome assemblies. Heterochromatic regions found in the centromere and telomere structures, and TEs, represent particular repeated nucleotide sequences, vital for genome plasticity and function. If extensive contig collections are available as the result of *de novo* assembly starting from large ONT read data, sequence redundancy and the overrepresentation of repetitive elements are expected to impact the assembly size. The practicality of such assemblies is often questioned when subsequent structural annotations are intended. With regard to such challenges, we also propose an original, automatized, digital scaffolding strategy that uses bash and R scripts. Although this procedure is still under development, we demonstrate its use for filtering and ordering contigs, as well as pinpointing repetitive genomic regions, based on nucleotide queries corresponding to the reference genome genes or scaffold-extracted sequences.

## 2. Materials and Methods

### 2.1. D. suzukii Collection

*D. suzukii* individuals were caught in August 2022 in two locations within Bucharest, Romania, namely, the “Dimitrie Brândză” Botanical Garden (44°26′19.6″ N 26°03′43.2″ E) and the Research Development Institute for Plant Protection, or ICDPP (44°30′07.0″ N 26°04′11.4″ E). Baited traps were used in order to obtain viable individuals, using an adapted version of the design proposed by Lasa et al. [20]. In short, red polyethylene cups adorned with a black stripe made of electrical tape were pierced with 14 to 16 holes of roughly 3.5 mm diameter evenly disposed in two rows, one spanning the black stripe and the other underneath it. Within each cup we dispensed banana–yeast–agar–semolina based standard culture medium and insert an additional capped tube device containing a supplementary attractant. The tube lid was punctured with 7–8 small orifices that allowed the passing of attractant volatiles. The extra attractant solution was based on 10% alcohol apple cider vinegar (ACV), into which we added approximately the same volume of a sugar–yeast mixture obtained by mixing 40 mL distilled water, 2.6 g sugar, and 0.42 g dry yeast (Pakmaya, Rompak SRL, Pascani, Romania). The traps were covered with cling film (Fino, Sarantis Romania S.A., Bucharest, Romania) and then placed hanging from tree branches with the use of yarn knots and loops.

For simplicity, we symbolized the lines collected from Botanical Garden and ICDPP as GB-ls-coga4 and ICDPP-ams-1, respectively. The GB-ls-coga4 line started from a few males and females collected during the first days of August 2022, and the ICDPP-ams-1 line started from two males and two females recovered on 1 August 2022.

### 2.2. D. suzukii Husbandry

Individuals collected from the two locations were placed in glass vials containing the standard culture medium. Our recipe is highly similar to the one presented at https://www.protocols.io/workspaces/drosophoto.com/resources/fruit-fly-food-banana (accessed on 17 December 2024), but instead of malt, molasses and nipagin we used similar quantities of semolina, refined sugar and propionic acid, respectively. The GB-ls-coga4 and ICDPP-ams-1 lines were maintained at a constant temperature of 21–23 °C and in alternative light–dark cycles, either approximately 12 h:12 h or natural (44°26′22.96″ N, 26°3′53.42″ E). The adult individuals were transferred to new vials every 2–3 weeks due to the slightly longer life cycle and somewhat weak adaptation to the artificial culture medium of *D. suzukii* in comparison to *D. melanogaster*. We observed that maintaining the lines at 18–19 °C had a negative impact on development and survivability, with the preferred temperature being around 23 °C. Prior to genomic sequencing, both *D. suzukii* lines were kept in laboratory settings over a year; thus, we expected them to resemble a near isogenic line, due to having a relatively homogenous genomic background, especially regarding the allelic diversity.

### 2.3. DNA Extraction

Individuals from the two lines were subjected to two separate DNA extractions performed with the Monarch HMW DNA Extraction Kit for Tissue (New England Biolabs, Ipswich, MA, USA), according to the recommended protocol. Firstly, we started from 25 males and proceeded with a 1400 RPM agitation speed step at 56 °C on a TS-100 Thermo-Shaker (Biosan, Riga, Latvia). For the second DNA extraction, 40 individuals (20 males and 20 females) were selected, but during the agitation step the speed was lowered to 700 RPM. DNA concentrations were measured with the ASP-2680 spectrophotometer (ACTGene, Piscataway, NJ, USA).

### 2.4. Whole Genome Sequencing

The sequencing was performed according to ONT long-read sequencing protocols that are using for the library preparation with the SQK-LSK114 Ligation Sequencing Kit V14 (ONT, Oxford, UK). At the corresponding step of the protocol, we employed the Long Fragment Buffer when processing the DNA extracted by using a 700 RPM agitation speed, or else the Short Fragment Buffer.

The DNA concentrations were repeatedly measured during library preparation using the Quantus Fluorometer (Promega, Madison, WI, USA) and the QuantiFluor ONE dsDNA System (Promega, Madison, WI, USA). The final concentrations of the DNA libraries and an approximate fragment length of 40 kb was considered for calculating the DNA sequencing input as femtomoles (fmoles) using an online conversion tool (https://www.bioline.com/media/calculator/01_07.html, accessed on 8 September 2023).

The DNA libraries were loaded on either MinION Flow-Cell R10 (two in total, one for each line) or Flongle Flow-Cell R10 (ten in total, five for each line) in a MinION MK1B device (all from ONT, Oxford, UK) coupled with the Flongle adaptor when appropriate. For simplicity, we will use the formal product codes for the MinION and Flongle flow-cells, namely, FLO-MIN114 (ONT, Oxford, UK) and FLO-FLG114 (ONT, Oxford, UK), respectively. During the preparation of the flow-cell priming mix, we added either 5 µL (FLO-MIN114) or 1 µL (FLO-FLG114) of 10 mg/mL Bovine Serum Albumin (Promega, Madison, WI, USA). The sequencing reaction lasted for at least 48 h on FLO-MIN114 (ONT, Oxford, UK) and 24 h on FLO-FLG114 (ONT, Oxford, UK).

The sequencing experiments were performed using the CPU version of the MinKNOW software (version ≥ 23.04.6) on Linux or Windows operating systems. During the sequencing runs we opted for not performing active basecalling, barcoding, or alignment, and the output file format was set to POD5.

### 2.5. Sequencing Reads Analysis

Basecalling of POD5 data was performed with the dedicated Dorado software (versions 0.5.0 or alternatively version 0.5.2). DNA specific basecalling models, namely, dna_r10.4.1_e8.2_400bps_fast@v4.3.0 (for short, fast) or dna_r10.4.1_e8.2_400bps_sup@v4.3.0 (for short, sup), were implemented using either the CPU- or GPU-based systems. The fast-basecalling model offers lower processing time at the expense of read quality, while sup-basecalling is computationally more intensive but usually provides better overall read quality.

The number, quality, length, N50 value, and data distributions of basecalled reads in FASTQ format were evaluated with NanoPlot (version 1.42.0).

### 2.6. Genome Assembly

The read libraries corresponding to GB-ls-coga4 and the ICDPP-ams-1 lines were assembled with the Canu software (version 2.0 or 2.2) on a Linux-based workstation equipped with two IntelXeon Gold 5418Y CPUs, 256 GB DDR4 RAM, and at least 8 TB of HDD storage. The assembly command contained several customized values, which included arguments such as GenomeSize = 260 m, minReadLength = 1000, minOverlapLength = 100 (or 500), maxMemory = 256, and maxThreads = 96.

For each line, the initial Canu assemblies exclusively exploited the FLO-MIN114 data handled with both fast (denoted GB-ls-coga4_fast and ICDPP-ams-1_fast, respectively) and sup (denoted as GB-ls-coga4_sup and ICDPP-ams-1_sup, respectively) basecalling models. One such assembly for each line was submitted to GenBank/NCBI (GB-ls-coga4_fast—GCA_037074645.1; ICDPP-ams-1_sup—GCA_040114545.1). For the GB-ls-coga4 line only, we generated additional assemblies by using FLO-MIN114 reads together with unaltered or split FLO-FLG114 reads, all basecalled with the sup model (denoted GB-ls-coga4_all and GB-ls-coga4_polca, respectively). The FLO-FLG114 reads were split evenly in 250 base non-overlapped k-mers using an original bash script (https://github.com/DL-UB/FastqSequenceSplitter, accessed on 1 June 2024) in order to implement an adapted polishing strategy with POLCA [21], which was originally tailored to make use of additional short reads.

The quality of the GB-ls-coga4 assemblies was assessed using QUAST version 5.2.0 [22] and BUSCO version 5.6.1 [23]. We ran QUAST to compare the tested Canu assemblies to the publicly available *D. suzukii* reference genome, Dsuz_RU_1.0 (NCBI RefSeq assembly GCF_037355615.1), with an estimated size of 178.7 Mb, and its corresponding gene annotations. The BUSCO command made use of the diptera_odb10 database as a reference.

### 2.7. Scaffolding

Various scaffolding procedures on GB-ls-coga4 genome assembly were performed with the original Digital Scaffolding (dScaff) procedure. In short, using BLAST data gathered by aligning the query sequences obtained from reference or publicly available assemblies of a given species against a collection of contigs, dScaff is able to order and map the respective contigs relative to the chromosomes or scaffolds harboring the query sequences. The dScaff workflow was developed in the *Drosophila* Laboratory, Faculty of Biology, University of Bucharest, as open-source. The package is freely available on GitHub at https://github.com/DL-UB/dScaff (accessed on 30 October 2024). The algorithm implemented in dScaff and details regarding its performance are presented elsewhere [24].

To generate graphical visualization intended to expose qualitative differences between various assembly and scaffolding procedures, we employed gene and ranked-queries particular to the LBDM_Dsuz_2.1.pri assembly (GenBank GCA_013340165.1, NCBI RefSeq GCF_013340165.1). Although the current reference genome is Dsuz_RU_1.0, we chose the previous reference assembly because we specifically wanted to test if the dScaff procedure can also successfully accommodate scaffold-level assemblies. In addition, we ran dScaff together with lists of Dsuz_RU_1.0 chromosome-specific ranked-queries to generate corresponding chromosome-level minimal complete scaffolds made of a selection of ordered contigs pertaining to GB-ls-coga4_all assembly (GenBank JAYGJW000000000). In order to extract regularly spaced genomic sub-sequences, we implemented an original open-source bash routine available on GitHub at https://github.com/DL-UB/SubSequencesExtractor (accessed on 20 October 2024). We configured dScaff to exploit either all gene sequences within the reference scaffolds that were longer than 2500 nucleotides or corresponding ranked-queries, and included in the final assortment only the BLAST [25,26] alignments complying with the standard successive thresholds of gene or ranked-queries coverage [24]. For the ranked-queries method, we considered genomic fragments of 4000 nucleotides, followed by a spacing of either 5000 nucleotides for a selection of scaffolds pertaining to the 2R chromosome from the LBDM_Dsuz_2.1.pri assembly or 20,000 nucleotides when exploiting the chromosomes from the Dsuz_RU_1.0 reference assembly.

The scaffolding implementation uses NCBI-BLAST+ version 2.12.0 [25], git version 2.34.1 [27] and seqtk 1.3-r106 [28] Linux packages, R version 4.1.2 [29] and dplyr version 1.1.4 [30], readr version 2.1.5 [31], utils version 2.12.3 [29] and tibble version 3.2.1 [32], and ggrepel version 0.9.5 [33] and scales version 1.3.0 [34] R packages. Scaffolding was performed in Linux Mint 21.2 Cinnamon coupled with a i9-13850HX processor (Intel, Santa Clara, California, USA), 32 GB DDR5 RAM, and 1 TB of storage.

### 2.8. Statistical Analysis

Statistical implementations were performed in R version 4.3.0 [29] and RStudio IDE [35]. The paired two-sample Student’s *t* tests were conducted using the *t.test* function as part of the stats package [29] by customizing *paired = TRUE* and *var.equal = FALSE* arguments. The two-way ANOVA with repeated measures tests were implemented using either the *aov* function from the stats package [29] or the *anova_test* function from the rstatix package [36]. Data manipulations were conducted with the dplyr package [30], and graphical representations were created with the ggpubr [37] and ggplot2 [38] packages.

## 3. Results

### 3.1. The Collection of D. suzukii Individuals

Using the custom fly traps, we successfully gathered *D. suzukii* individuals from two locations in Bucharest (Figure 1). From the Botanical Garden and ICDPP we collected adult individuals found in several traps, and corresponding cultures were successfully stabilized. The collection sites are several kilometers apart, both within the urban area of Bucharest, with the Botanical Garden being a public place and the specific cultivated area of ICDPP being usually accessible only to its employees.

### 3.2. Sequencing Experiment

Extracted DNA concentrations corresponding to the two experimental *D. suzukii* lines were evaluated by spectrophotometric measurement. Using the protocol involving a 1400 RPM agitation speed, for the GB-ls-coga4 and ICDPP-ams-1 lines we obtained DNA concentrations of 62.1 ng/µL and 56.2 ng/µL, respectively. As expected, we observed a higher DNA concentration when using more individuals (40 compared to 25) and the 700 RPM agitation speed, namely, 160.6 ng/µL for the GB-ls-coga4 line and 83.4 ng/µL for ICDPP-ams-1.

The DNA samples were further used for library preparation and subsequent sequencing on either FLO-MIN114 or FLO-FLG114. We opted to use DNA libraries with higher fmole concentrations on FLO-MIN114 for each experimental line. In the case of GB-ls-coga4, Table 1 summarizes the DNA input concentrations, the flow-cell type, and the corresponding sequencing data for each run based on sequencing reports generated by MinKNOW.

Referring to the total number of bases, the total output from the five FLO-FLG114 flow-cells (1.274 Gb harbored by 0.608 M reads) represented 7.68% of the sequencing data gathered by using FLO-MIN114. On average, each FLO-FLG114 had around 50 pores available for sequencing; thus, the total number of pores harbored by all of them was roughly 250, which represented 16.5% of the total pores available for our FLO-MIN114. The two top-performing FLO-FLG114 contributed more than 50% to the data generated by all FLO-FLG114 flow-cells (or 4.07% of FLO-MIN114 data). Similar DNA inputs and ensuing sequencing outputs were generated for the ICDPP-ams-1 line. Since the same specific analysis procedure was applied for evaluating the sequencing and genome assembly results, we will hereafter present and discuss mainly the data associated with GB-ls-coga4 in order to assure a more focused analysis. However, the complete data are available in Appendix A.

Several one-sample *t*-tests were performed on the sequencing parameters presented in Table 1 in order to examine the differences between FLO-FLG114 and FLO-MIN114 flow-cells. In summary, the values recorded for the five FLO-FLG114 were compared with the single value corresponding to the FLO-MIN114, which served as the reference (the complete results are presented in Appendix A). Regarding the input DNA concentration and N50 values, we did not find differences that were statistically significant. In contrast, the output file size was significantly different between the two types of flow-cells (t = −353.583, *p* = 3.84 × 10^−10^, sample mean = 2.98, reference mean = 198.7). The read counts and number of bases variables were log_2_ transformed because of the one-order-of-magnitude differences between them, and consequently, we identified statistically significant differences for both of them (reads count: t = −11.914, *p* = 2.843 × 10^−4^, sample mean = 16.18, reference mean = 22.133; number of bases: t = −16.99, *p* = 7.037 × 10^−5^, sample mean = 27.779, reference mean = 33.949). The complete results obtained by running one-sample *t*-tests are presented in Appendix A.

Sequencing reads quality was evaluated using NanoPlot, following Dorado basecalling with either fast or sup models. Detailed data are presented in Appendix A, including mean and median read length, mean and median read quality, and percentages of reads with Phred Q scores higher than thresholds of 7, 10, or 15, calculated from the total number of reads in each sequencing reaction. Given the significant differences in sequencing output features between the two types of flow-cells, we aimed to assess eventual differences concerning the quality parameters available for the reads. One-sample *t*-tests were separately performed on fast and sup-basecalled data by comparing the values corresponding to the five FLO-FLG114 to the single value per parameter recorded for FLO-MIN114.

Concerning GB-ls-coga4, the statistical analysis provided similar results regardless of the basecalling model. More precisely, we calculated significant differences for mean/median read length and the log_2_ transformed number of actual reads, achieving the selected quality score thresholds of 7, 10, and 15 (Appendix A). When evaluating the mean read quality, we found no significant differences, although the values corresponding to FLO-FLG114 were slightly higher. The median read quality was similar for the fast-basecalled reads, but it was significantly different for the sup-basecalled reads. The log_2_ transformed number of reads revealed statistically significant differences between data obtained with FLO-FLG114 and FLO-MIN114 for all quality thresholds, regardless of the basecalling model.

Additionally, we made use of data presented in Appendix A in order to compare the impact of Dorado basecalling models on the quality parameters evaluated for the reads by using paired two-sample Student’s *t*-test and two-way ANOVA with repeated measures. Except for the mean/median read quality, where the actual values were handled, for all the remaining parameters we performed the statistical analyses on both actual and log_2_ transformed data (Appendix A). In accordance with the intuitive assessment of data, we found that the differences generated by the two basecalling methods were statistically significant. When comparing the log_2_ transformed number of reads reaching the three considered Phred Q scores, we also found significant differences between fast and sup-basecalling methods, especially in the case of reads with a Phred Q score of over 15 (Figure 2).

Moreover, we compared the three sets of reads corresponding to the three Phred Q score thresholds by running two-way ANOVA with repeated measures on the log_2_ transformed values and found statistically significant results. The two-way ANOVA investigated the differences between reads obtained with the two basecalling models, as well as for the three quality thresholds. The differences were more evident when using the anova_test function (F = 714.189, *p* = 4.11 × 10^−6^ between the basecalling models; and F = 857.867, *p* = 7.8 × 10^−7^ between the Phred Q score thresholds) compared to the aov function (F = 8.065, *p* = 0.009 between the basecalling models; and F = 6.588, *p* = 0.005 between the Phred Q score thresholds).

Similar results were obtained when analyzing read data collected for the ICDPP-ams-1 sequencing experiment, both regarding the paired two-sample Student’s *t* tests (Appendix A) and two-way ANOVA with repeated measures.

### 3.3. Quality Assessment of the Genome Assemblies

In our study, we evaluated the potential impacts on genome assembly output of several features that are expected to be decisive for such procedures, including read quality, starting data quantity, and an alternative strategy of using long-read sequencing data. Therefore, we primarily used Canu to generate a series of genome assemblies with data collected for GB-ls-coga4. Specifically, we uniquely exploited the fast and sup-basecalled reads generated by FLO-MIN114, as well as the sup FLO-MIN114 reads together with either unaltered or split FLO-FLG114 sup-basecalled reads. The resulting assemblies were denoted GB-ls-coga4_fast, GB-ls-coga4_sup, GB-ls-coga4_all, and GB-ls-coga4_polca, respectively. The latter assembly was based on GB-ls-coga4_sup assembly further enriched with a collection of 250 nucleotides long sequences that resulted after evenly splitting the FLO-FLG114 reads in the respective non-overlapped k-mers. Subsequently, the assembly and the processed reads were submitted to a polishing-like strategy via POLCA.

The quality of the four assemblies was evaluated using QUAST [22] and BUSCO [23]. As shown in Table 2, the QUAST output indicates a 3.11% increase in genome fraction coverage when using the sup-basecalling model compared to the fast-basecalling one. However, the genomic fractions of GB-ls-coga4_all and GB-ls-coga4_polca were decreased by 1.879% and 0.135%, respectively, compared to GB-ls-coga4_sup. These relatively small changes in the genome fraction coverages were accompanied by an increase in duplication ratio, rising from 1.497 in GB-ls-coga4_fast to 1.928 in GB-ls-coga4_sup. Notably, the GB-ls-coga4_all had a higher genome fraction value than GB-ls-coga4_fast, and its duplication ratio was lower, at 1.378.

Switching to the sup-basecalling model also resulted in a more than two-fold increase in the number of contigs compared with GB-ls-coga4_fast, rising from 2344 to 4793. GB-ls-coga4_polca comprised the same number of contigs as the GB-ls-coga4_sup, as the polishing step had a minimum impact on contig statistics. Interestingly, GB-ls-coga4_all, which was generated by using the largest number of long-reads, contained the fewest contigs, but it harbored the bigger N50 value. The longest contig was unique to GB-ls-coga4_sup, and we noticed a significant value drop in this parameter in GB-ls-coga4_all and a massive four-fold reduction of it in GB-ls-coga4_fast.

The 0.608 M reads generated using the FLO-FLG114 did not enhanced the genome fraction metric. However, notable improvements were observed in the duplication ratio, which decreased from 1.928 to 1.378 when incorporated into a new assembly. In contrast, its application during the polishing stage resulted in only a marginal difference of 0.004, but the main expected improvement was expected to be related to the overall nucleotide sequence quality and accuracy. Although this aspect was not thoroughly investigated, it is conceivable that using Canu 2.0 for generating GB-ls-coga4_sup and Canu 2.2 for GB-ls-coga4_all could impact some of the properties particular to these two genome assemblies.

We also implemented a complementary quality evaluation using BUSCO with the Diptera lineage-specific dataset to assess the completeness and redundancy of the four assemblies (Appendix A). As detailed in Table 3, using the sup-basecalling model resulted in a marked increase in assembly completeness, rising from 84.5% in GB-ls-coga4_fast to 98.6% in the GB-ls-coga4_sup. Nevertheless, this improvement was accompanied by a notable increase in the proportion of duplicated genes, which rose from 12.9% to 32.2%. Conversely, the percentage of fragmented genes significantly decreased, from 9.1% to 0.8%, as did the percentage of missing genes, which dropped from 6.4% to 0.6%.

Completing GB-ls-coga4_sup with the collection of FLO-FLG114 long-reads did not result in a substantial enhancement in complete, fragmented, or missing gene metrics, but significant improvements were observed regarding both the single-copy genes that could be identified and the duplication rate. More precisely, the single-copy metric increased by almost 23%, while the percentage of duplicated genes decreased from 32.2% to 9.5% in GB-ls-coga4_all.

The polishing-like procedure using sub-sequences extracted from the FLO-FLG114-generated long-reads did not yield remarkable improvements in the assembly quality scores inferred by BUSCO when comparisons with GB-ls-coga4_sup were considered.

Because of genomic sequence redundancy, GB-ls-coga4_all had inferior BUSCO scores on single-copy and duplicated statistics compared to the reference sequence (Dsuz_RU_1.0_all) but was better on all the remaining ones.

### 3.4. Assembly Annotations Using dScaff Procedure

We tested for eventual nucleotide sequence quality differences between the GB-ls-coga4_sup, GB-ls-coga4_all, and GB-ls-coga4_polca assemblies by submitting them to the dScaff procedure with gene or ranked-queries lists particular to the NW_023496808.1 scaffold from the 2R chromosome of the LBDM_Dsuz_2.1.pri assembly. Thenceforth, we compared the length and the mismatch/gap content of the specific BLAST alignments generated between the assemblies and the considered queries. To control for the non-specific alignments spawn by repetitive sequences, we filtered out redundant alignments for any query and subject pair by considering only unique queries for any subject sequence. For example, if a subject sequence created multiple alignments with different queries, we considered one alignment for each query.

Firstly, we examined data generated with the gene-queries, and statistical analysis revealed no significant difference between the alignments lengths and the specific content of mismatches or gaps. Similar results were obtained when using both the one-way ANOVA test with Tukey’s Honestly Significant Difference (HSD) post-hoc test and the Kruskal–Wallis one-way ANOVA with Dunn’s Multiple Comparison post-hoc test. In the same way, when implementing the ranked-queries strategy, we observed no differences when evaluating the lengths, mismatches, or gaps of the BLAST alignments. Although not statistically significant when α = 0.05, a noticeable result was identified when analyzing the alignments lengths for ranked-queries, with F = 2.483 and *p* = 0.084 for ANOVA investigation. This result was supported by Tukey HSD tests, which showed close to significance differences between the GB-ls-coga4_sup and GB-ls-coge4_all assemblies (*p* = 0.095), as well as between GB-ls-coga4_all and GB-ls-coga4_polca (*p* = 0.139). In contrast, Kruskal–Wallis one-way ANOVA and Dunn’s Multiple Comparison post-hoc tests revealed no noticeable differences between assemblies.

The “mapped_contigs.csv” file generated by dScaff contains tabular data, where each alignment of interest between a given contig (represented on *Y*-axis) and one or more query sequences (represented on *X*-axis) is indicated in a distinctively marked cell, herein called a positive cell. The other cells are unmarked. By simply coloring the positive cells, it was possible to obtain a suggestive graphical representation featuring the ordered arrangement of all specific contigs and various non-specific hits localized within the envisioned chromosome/scaffold. We considered that in order to facilitate certain analyses and annotations, it was of interest to attain a selection of contigs assuring a simple or minimal linear coverage of a chromosome or of a specific genomic sub-region (a so-called minimal complete scaffold). In comparison with a finished and accurate reference scaffold, such a minimal scaffold does not provide the contig orientations relative to a given DNA strand, and the coverage for the most part of the chromosome/genomic region is granted by single contigs, not consensus sequences. We opted for using the red color to depict the set of contigs pertaining to such a minimal complete scaffold and blue to show the positive cells corresponding to putatively redundant contigs or repeats-containing contigs. In Figure 3, we show the manually annotated representation of a subset of contigs pertaining to the GB-ls-coga4_sup and GB-ls-coga4_all assemblies covering the same genomic region from the NW_023496808.1 publicly available scaffold. Data were obtained using the gene-queries strategy.

According to our estimation, the addition of FLO-FLG114 data to the FLO-MIN114 read collection enabled Canu to produce an output with fewer redundant contigs (Figure 3B) than the GB-ls-coga4_sup assembly (Figure 3A). The differences in terms of red contigs content were limited.

To pinpoint the regions rich in repeats within the chromosome/scaffold, we found that using the ranked-queries strategy was the most effective approach. To reach this conclusion, we made use of the same NW_023496808.1 scaffold as the reference, and both gene and ranked-queries stratagems were applied on GB-ls-coga4_all. Our results show that for a given genomic region, the ranked-queries methodology provided more valuable insights than the alternative (Figure 4). The candidate contigs that made the minimal complete scaffold were roughly comparable when either scaffolding strategy was operated, but the information regarding the repetitive nature of a certain genomic region was more vividly apparent when genomic sequences outside the genes were also considered as queries (Figure 4B). In addition, we also discovered that when particular reference scaffolds were highly enriched with genomic repeats, the ranked-queries tactic outcompeted the gene-queries one when the identification of the contigs assuring the genomic sequence linear coverage was pursued (Appendix A).

The previous analyses made use of all the filtered positive BLAST hits that were included only within the “mapped_contigs.csv” file, which had to be manually handled in order to color the alignments established between the contigs and the query lists.

Both BUSCO and dScaff comparisons showed relatively elevated percentages of redundancy even within GB-ls-coga4_all assembly, which we found to exhibit the best characteristics. In this context, we decided to use dScaff to automatically generate the complete set of chromosome-level minimal complete scaffolds corresponding to this particular genomic assembly. Such a minimal assembly can be extremely useful if rapid gene or transposon screenings or annotations are intended. For this specific endeavor, we obtained the specific lists of ranked-queries corresponding to the X, 2L, 2R, 3L, 3R, and 4 chromosomes from the Dsuz_RU_1.0 reference assembly. Currently, this reference assembly does not contain Y or mitochondrial chromosomes but has a series of 28 unallocated scaffolds that we chose not to use for query list generation. Therefore, although the complete GB-ls-coga4_all assembly is expected to harbor both Y and mitochondrial contigs, we specifically pursued only the contigs containing genomic sequences corresponding to the fully assembled reference chromosomes. The resulting output consisted of a selection of ordered contigs grouped and mapped according to their chromosome or chromosome arm origin, here denoted as GB-ls-coga4_min. In Figure 5, we present the automatic graphical output compiled for the 2R chromosome, where only the 22 red contigs ensuring the basic linear coverage are represented. According to dScaff estimations, at least 90% of the total length of the 2R chromosome was linearly covered by the selected contigs. Overall, the complete genome scaffolding allowed for the selection of a total of 84 contigs, which represented a genome fraction of about 89%, according to QUAST estimations.

In order to make a primary quality assessment of this minimal assembly, GB-ls-coga4_min, we ran BUSCO on both our data and a variant of the reference assembly containing only the complete chromosomes that actually provided the template for the queries used by dScaff. The results are presented in Table 4.

## 4. Discussion

The modern journey of *D. suzukii* started in Japan in the first half of the 20th century, and it took nearly 50 years for this insect to reach the United States of America, where it was first reported in Hawaii, then, several years after, in California [39,40]. Roughly around the same period, *D. suzukii* was captured in Spain, and during the following six years, its presence was recorded all around Europe [1]. In Romania, it was first reported in the northern region of Bucharest [41], and afterwards its presence was registered only in the Oltenia region [42], both located in the southern portion of the country. None of the previous local endeavors aimed to establish a line of *D. suzukii* and perform subsequent analysis, but their focus was solely on reporting this insect pest and making some evaluations regarding its potential danger for the agriculture sector.

We collected the starting individuals from two sites within Bucharest, namely, ICDPP, where *D. suzukii* was previously noticed, and GB, a location hosting a vast variety of biocenoses. From the start, our aim was to sequence, assemble, and annotate the two genotypes, a needed enterprise especially considering that at the date of our collections, in August 2022, there were only three assemblies reported in the NCBI, two over a decade old and the former reference. In the meantime, besides the assemblies submitted by our group, just two more chromosome-level genomic assemblies were indexed in the NCBI, namely, the current reference from Rockefeller University and one from the University of Montpellier. Except for our sequencing projects, the other five reported assemblies used either a hybrid approach, mixing Illumina NGS and PacBio data, or only sequencing data generated by one of these two methodologies.

Considering the aforementioned context, our genomics project establishes several firsts, such as the first genome sequencing of a *D. suzukii* population from Eastern Europe, the first time when exclusively ONT data are used for the genome assembly of this species, and the first time when various stratagems for bettering FLO-MIN114 with the aid of FLO-FLG114 data are undertaken.

The ONT sequencing method is gaining increasing popularity, especially due to its non-pretentious infrastructure and reasonable affordability when it comes to reactive and flow-cell costs. Our past experience made us aware of the somewhat low reusability of FLO-MIN114; therefore, we intended to test a cost-effective alternative by using the FLO-FLG114 system. In doing so, we hoped for a valuable increase in volume of sequencing data that could eventually significantly improve the assembly quality.

For the sequencing of GB-ls-coga4, we used both FLO-MIN114 and FLO-FLG114, thus generating over 17.8 billion bases comprised in approximately 5.2 million reads; data analysis revealed major output differences between the two flow-cell types. Even if the mean DNA input for FLO-FLG114 (14.962 fmoles) was similar to the one for FLO-MIN114 (16.18 fmoles), the latter represented only 17.8% of the overall input DNA, but it generated over 92.8% of total bases harbored by 88.3% of the produced reads. Although there was a clear difference regarding the total number of pores available for sequencing, this result is intriguing, because if similar per-pore outputs are to be expected regardless of the type of flow-cell, then the actual FLO-FLG114 output should be larger. Actually, the FLO-FLG114 achieved an average of 5.1 Mb per single pore, less than half of the mean number of bases produced by a single pore from FLO-MIN114. It is tempting to ascribe this discrepancy to the differences in flow-cells design, but ONT methodologies were reported to generate highly different bases and reads counts outputs both for FLO-MIN114 [12,42,43,44] and for FLO-FLG114 [11,45,46,47,48]. Accordingly, compared with the output particularities observed for GB-ls-coga4, the data obtained for ICDPP-ams-1 showed only slightly elevated per-pore output data in favor of the respective FLO-MIN114.

In fact, by using FLO-FLG114, we obtained about 0.608 M reads, representing 7.68% of the data generated with FLO-MIN114. However, based on our results, sequencing with FLO-FLG114 led to a variety of data outputs that appeared inconsistent, a feature for which we do not have a plausible explanation. For instance, in two runs using the same DNA input of 20 fmoles, we observed considerable variation: 34,940 reads were obtained with 49 pores, whereas 155,390 reads were obtained with 43 pores.

The N50 statistics were similar between the two types of flow-cells; therefore, assuming that the differences spotted in the case of GB-ls-coga4 are an exception, one can consider FLO-FLG114 for adding data to the more proficient FLO-MIN114. It is worth mentioning that if such an alternative is sought after, in order to be cost effective, the average number of starting available pores for FLO-FLG114 should be at least 100. Unfortunately, in our experience, we never observed that many starting pores, a feature that, if also encountered by many other research groups, should be thoroughly addressed by ONT.

The full extent of our sequencing data is comparable with previous accounts. As an example, a 2020 study on *Panthera leo* genome sequencing used the now-worn-out FLO-MIN106D (R9.4.1), generating 1.5 M reads over 48 h [49]. In contrast, our single FLO-MIN114 employed for GB-ls-coga4 yielded 4.6 M reads, which represents a significant increase in throughput. Similarly, a study on *Pavo cristatus* [50] using FLO-MIN106D reported only 366,323 reads after 48 h of sequencing, another argument in favor of the latest R10 chemistry, which allows for a substantial improvement in read output. In addition, it is possible that not only the data output but also the read accuracy and methylation calling to be significantly improved compared with the previous flow-cell versions, as argued by Ni et al. [51]. Regarding data yield by using FLO-FLG114, a study aiming to achieve pathogen identification and resistance gene detection by using the outdated FLO-FLG001 (R9.4.1) gathered 93,892 reads after a 24 h run using 24 pores [46], better than one of our FLO-FLG114 flow-cells, which generated 34,940 reads using 49 pores. Conversely, their highest output of 293,841 reads was achieved with 62 pores, whereas a FLO-FLG114 harboring 60 pores produced 224,540 reads particular to GB-ls-coga4, a comparable feat. Another study focused on enterovirus genotyping in diagnostic samples conducted two sequencing runs with FLO-FLG001 containing 27 and 42 pores, yielding 190,290 reads with a base count of 135.55 Mb and 267,380 reads with a base count of 172.4 Mb. We also had one FLO-FLG114 with 43 pores that produced 155,390 reads but with a higher base count of 268.84 Mb, indicating longer average read lengths.

ONT sequencing data can be converted into nucleotide reads by using several basecalling methods, the most rapid and least qualitative of them being the fast model, while the sup model supposedly offers the most accurate reads but requires higher computing power and substantially increased processing time. We wanted to test both of these options in order to evaluate their impact over the reads and, subsequently, the assembly quality. As expected from the explanation provided by the developer, the sup-basecalling model generated better results compared to the fast one with regard to the mean read length, median read length, mean read quality, median read quality, percentage of reads with Q > 7, percentage of reads with Q > 10, percentage of reads with Q > 15, log_2_ number of reads with Q > 7, log_2_ number of reads with Q > 10, and log_2_ number of reads with Q > 15. The superior performance of the sup-basecalling model was previously documented [52,53,54].

When the sup-basecalling model was considered, the analysis of reads quality corresponding to the FLO-MIN114 and FLO-FLG114 indicated that quality parameters differed significantly between the flow-cell types. With the exception of mean read quality and median read quality, all investigated parameters were significantly better for reads generated by FLO-MIN114 compared to FLO-FLG114. Similar results were also obtained for the ICDPP-ams-1. The median read length and quality were arguably better parameters for describing the dataset than the mean, as we did not perform any length or quality filtering on read data, therefore keeping eventual outlier values.

Our results clearly indicate that the sup-basecalling model is the most efficient and provides higher-quality reads that are expected to have a discernable impact on the genomic assembly procedure, regardless of the specific assembler in use.

In view of the novelty of our ONT approach and the results concerning the quality of our sequencing data, we decided to test various types of genome assemblies. Thus, we decided that the starting data should be the fast and sup FLO-MIN114 reads, the set of sup reads gathered from both FLO-MIN114 and FLO-FLG114, as well as a particular set of sup reads resulting from combining FLO-MIN114 data with a particular makeover of the complete FLO-FLG114 output. These collections of reads led to a series of assemblies denoted GB-ls-coga4_fast, GB-ls-coga4_sup, GB-ls-coga4_all, and GB-ls-coga4_polca, respectively. The latter assembly resulting from the FLO-MIN114 data were adorned with 250 nucleotide-long k-mers uniquely extracted from each FLO-FLG114 read. We believed that such an approach would allow us to efficiently use our sequencing data together with the very well-established polishing procedures that regularly employ short-reads.

In order to evaluate the quality of the resulting assemblies, we employed two widely recognized software tools, namely, QUAST and BUSCO. As previously stated, QUAST provides information on assembly completeness relative to a reference genome, a report on assembly redundancy, as well as technical metrics such as the total number of contigs, the longest contig, and the N50 value. In our hands, QUAST evaluation identified GB-ls-coga4_all as the most promising assembly. Although the genome fraction parameter was slightly lower than that of the GB-ls-coga4_sup and GB-ls-coga4_polca, the duplication ratio parameter was significantly reduced in GB-ls-coga4_all, reflecting its higher quality. Furthermore, substantial differences were observed between the GB-ls-coga4_fast and GB-ls-coga4_sup assemblies, highlighting the pivotal role of the basecalling model in determining assembly quality.

BUSCO uses a collection of highly conserved genes to assess assembly completeness by providing carefully calculated values for key parameters, referring to the number of complete, single-copy, duplicated, fragmented, and missing genes. Alternatively, the set of considered genes can be also user defined, an approach that was previously described when a set of immunity-related genes was compiled and searched within the analyzed assemblies [55]. These results also confirm that GB-ls-coga4_all represents the best assembly, having the top characteristics for all the considered parameters. Moreover, we also ran BUSCO on the current reference assembly (here denoted as Dsuz_RU_1.0_all) and found that GB-ls-coga4_all scored better when the completeness of BUSCO data was primarily considered (complete, fragmented, and missing genes). Since GB-ls-coga4_all assembly consisted of an unfiltered set of contigs that were expected to exhibit a certain degree of redundancy, in comparison with the chromosome-level reference assembly it had fewer single-copy and more duplicated genes.

The addition of integral FLO-FLG114 reads to FLO-MIN114 data had a remarkable effect. More precisely, although the FLO-FLG114 data represented only approximately 7.2% of overall sequencing data, it contributed to a reduction in duplications of 30% and 70% according to QUAST and BUSCO inquiries, respectively. In addition, the important N50 contig statistic increased its value by a factor of almost 3.4× for GB-ls-coga4_all compared to GB-ls-coga4_sup.

To our surprise, using the FLO-FLG114 reads for polishing the GB-ls-coga4_sup had no discernable effect. This was apparent when QUAST and BUSCO procedures were employed, but also consecutive to other quality assessment methods that focused on comparatively evaluating the nucleotide sequences per se. During the process of dividing the FLO-FLG114 reads into 250 nucleotide-long k-mers, we generated a total of 5,573,157 k-mers, which were subsequently used for polishing with POLCA. Among these k-mers, homopolymers consisting of 10 consecutive adenine nucleotides were found in 74,930 k-mers, representing 1.34% of the total. Similarly, homopolymers of thymine with a length of 10 nucleotides were identified in 63,721 sequences, accounting for 1.14% of the total. In contrast, homopolymers of cytosine and guanine were much less frequent, appearing in only 0.08% and 0.09% of the k-mers, respectively. We consider that such repetitive structures have a greater weight in the context of the smaller 250-mers than when present within a significantly longer read and thus may influence the polishing process, resulting in trivial differences between the polished and the pre-polished assembly versions.

Several characteristics of the assemblies generated through different methodological approaches were addressed by QUAST and BUSCO, but we also intended to make use of alternative practices. Thus, we wanted to test for similarities and differences between the assemblies’ sequences and then obtain a visual representation of the contigs’ order within the chromosomal regions as well as other important features, such as contig redundancy and genomic regions enriched in repetitive sequences. For these purposes, we applied the original dScaff procedure in order to obtain relevant alignments between contigs and nucleotide queries corresponding to either gene or chromosome-/scaffold-extracted sequences. Inherently, dScaff retains extensive records of detailed BLAST alignment results that can be thoroughly analyzed. We opted to analyze only the assemblies based on sup reads, i.e., GB-ls-coga4_sup, GB-ls-coga4_all, and GB-ls-coga4_polca, by attempting to quantify the lengths, number of mismatches, or number of gaps found within the alignments obtained by mapping them to a standard sequence pertaining to the previous reference assembly. The scaffolding procedure was performed using either genes or ranked-queries collected from the NW_023496808.1 scaffold from the 2R chromosome. We found that in terms of length, number of mismatches, or number of gaps, the alignments generated by the three assemblies were highly similar, thus suggesting that they do not significantly differ. The strategy using ranked-queries generated more alignments to be analyzed, and this feature might be at least partially responsible for finding almost statistically significant differences when the alignment lengths were tested with ANOVA. It is possible that this similarity between the assemblies’ sequences is specific only to the NW_023496808.1 scaffold, but considering its dimension and structure particularities, we believe that it is truly representative of the entire genome. As previously noticed in the case of GB-ls-coga4_polca, the polishing procedure did not improve the quality of starting contig sequences in a conspicuous way. Even though the GB-ls-coga4_all had superior overall assembly quality parameters compared to GB-ls-coga4_sup, it seemed that the actual nucleotide sequences had somewhat similar attributes. This is something to be expected considering that both assemblies were generated by using sup-basecalled reads and shared about 88% of their start reads. Of course, it is also possible that our tactic of relying on evaluating BLAST alignments is not suitable for evaluating the quality of the nucleotide sequences pertaining to a collection of contigs.

In addition to facilitating a nucleotide sequence comparison between a selection of assemblies, dScaff also allowed us to portray the arrangement of contigs in agreement with a reference sequence as well as their putative overlaps with other concurrent contigs. A key output file supplied by dScaff is “mapped_contigs.csv”, which offers a plain visual representation of the alignments produced by chromosome-/scaffold-specific contigs and the corresponding queries. The file also displays several grouped or isolated additional alignments, which usually originate because a sub-sequence of a given query, often consisting of repeating units, can also be mapped in several other genomic locations. We chose to color in red the selection of contig alignments, assuring an abridged linear coverage of the scaffold, and color the remaining ones blue. By doing so, one can make an estimation of the genomic information redundancy exhibited by the collection of contigs pertaining to a given assembly. When comparing GB-ls-coga4_sup and GB-ls-coga4_all, covering the same genomic region from the NW_023496808.1 scaffold, in accordance with QUAST and BUSCO results, we clearly noticed less redundancy (fewer blue alignments) for the latter assembly. This comparison was carried out by using gene-queries; therefore, we tested whether, by using ranked-queries, which are vastly superior in number, the redundant nature of specific genomic region would be more apparent. Indeed, when both gene and ranked-queries strategies were applied on GB-ls-coga4_all and NW_023496808.1, the ranked-queries tactic allowed us to dramatically increase the “resolution” inside a certain genomic region pertaining to the reference scaffold, which is enriched in repetitive sequences. Such specific mapping strategies can be used in order to find candidate genomic regions that could harbor transposons and other types of repeats.

These analyses, made possible by dScaff, proved to be very useful for visualizing certain particularities of various assemblies. It also reinforced the results gathered with QUAST and BUSCO, showing that the total number of 1880 contigs pertaining to GB-ls-coga4_all contained redundant information that can be difficult to properly analyze in the context of specific genomic annotation processes. Thus, in order to make available a more supple assembly, we used dScaff to obtain a so-called minimal complete genomic assembly, which is actually a subcollection of 84 contigs from GB-ls-coga4_all that promise a continuous linear coverage of the *D. suzukii* reference genome. Since the resulting GB-ls-coga4_min assembly does not harbor data corresponding to Y and mitochondrial chromosomes, for BUSCO comparisons we used a compiled version of Dsuz_RU_1.0 containing only the completely assembled chromosomes. The results indicate that there were no noticeable differences between the two assemblies except for the number of duplicated genes, a feature at least partially determined by the overlapping regions between some of the adjacent contigs, such as tig00001190 and tig00001392, the largest contigs selected for 2R (see Figure 5).

The detailed analysis presented in this article reveals a strategy for obtaining the best genome assembly using various bioinformatics tools. The complete pipeline is presented in Figure 6.

## 5. Conclusions

The advent of highly advanced methods for gene and genome editing brought forth novel strategies for insect pest control, including *D. suzukii*. In order to successfully apply such approaches, one needs to rely on good-quality genome assemblies and annotations particular to the local pest populations that are to be subjected to preventive measures. The current methodological paradigm applied for genome sequencing considers the complementary production of both long-reads and short-reads due to their specific properties, such as the ability of long-reads to extend over repetitive genomic regions and the high quality of short-reads. As a consequence, projects intended to accomplish genome assembly, polishing, and annotations have to accommodate this particular assortment of sequencing data.

We presented here a genomic project involving the sequencing and assembly of the genomes pertaining to Romanian local *D. suzukii* populations located within the Bucharest city area. Our strategy made use exclusively of ONT sequencing data that were subsequently basecalled using both fast and sup models. As expected, a significant increase in reads quality was achieved consecutive to sup-basecalling, with the corresponding reads being further used for assembly purposes. When assembling the *D. suzukii* genomes, we tested the impact of a collection of FLO-FLG114 reads over the bulk sequencing data gathered with FLO-MIN114 by either plainly adding them to the latter reads or by using derived 250-mers for polishing procedures. According to our testing, the addition of supplementary reads, although representing a relatively small percentage of total sequencing data, had a critical effect on assembly metrics assessed by BUSCO and dScaff. The latter genomic tool was successful in exposing both the redundancy of the tested assemblies and the genomic subregions plausible to have a high content of repeats. It also allowed us to filter and rank the contigs pertaining to GB-ls-coga4_all in order to generate a considerable reduced assembly that offers a reliable linear genomic coverage and exhibits quality parameters on par with those particular to the reference genome.

In conclusion, our study presents a series of original methodological approaches based on ONT data that led us to obtain high-quality genome assemblies corresponding to Romanian populations of the *D. suzukii* insect pest.

## Figures and Tables

**Figure 1 insects-16-00002-f001:**
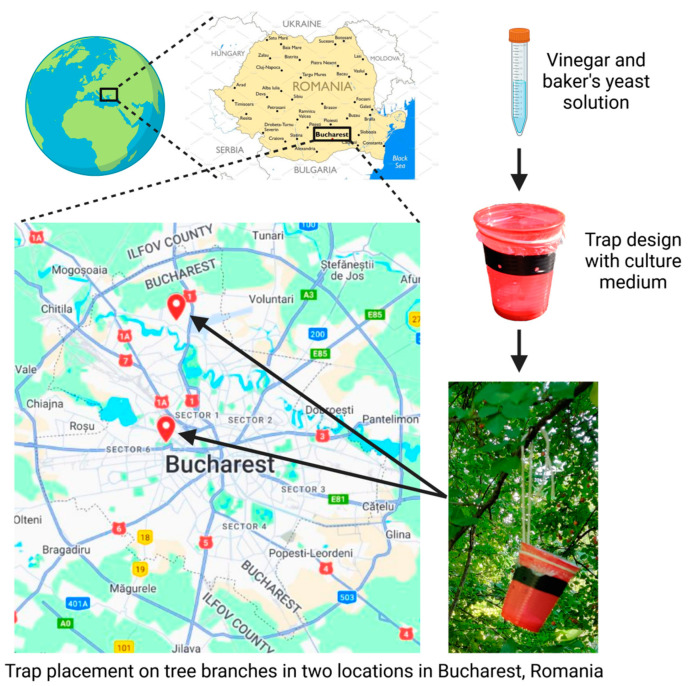
Trap design used for *D. suzukii* collection and the corresponding locations of trap placement. The two locations (upper, Research Development Institute for Plant Protection (ICDPP) at 44°30′07.0″ N 26°04′11.4″ E; lower, “Dimitrie Brândză” Botanical Garden at 44°26′19.6″ N 26°03′43.2″ E) are separated by various natural and artificial barriers, such as water flows and urban architecture. Created with BioRender.com (accessed on 23 December 2024).

**Figure 2 insects-16-00002-f002:**
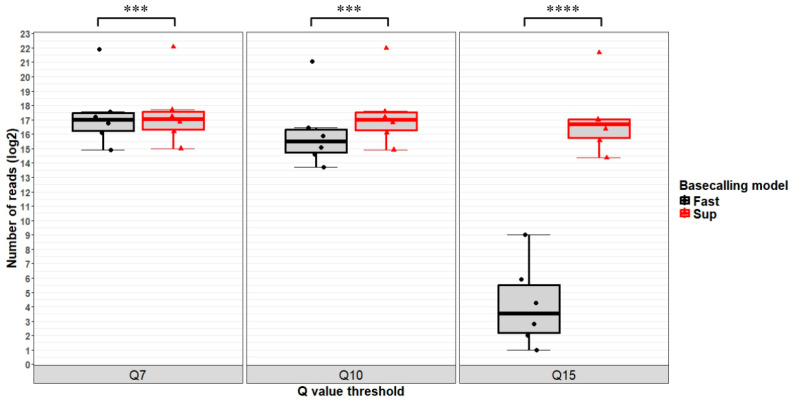
Boxplot of log_2_ transformed number of reads corresponding to the GB-ls-coga4 sequencing experiment with a Phred Q score higher than the threshold of 7, 10, or 15 obtained from either fast or sup-basecalling models. Statistical significance is shown as *** for *p* ≤ 0.001 or **** for *p* < 0.0001 based on paired two-sample Student’s *t* tests. Boxplots were created in R (version 4.3.0), and the collage was created with BioRender.com (accessed on 23 November 2024).

**Figure 3 insects-16-00002-f003:**
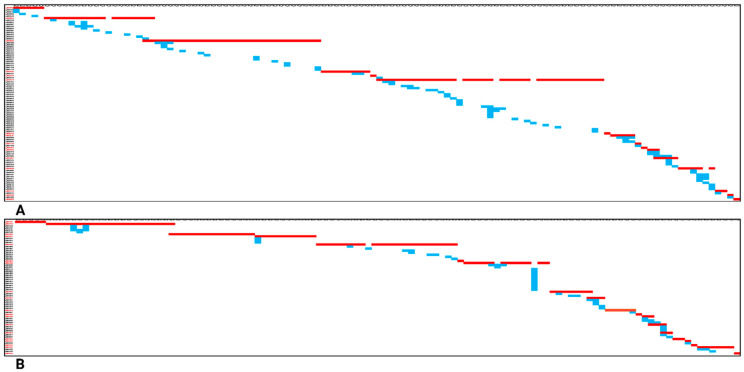
The figure represents the selection of contigs that were mapped on the NW_023496808.1 scaffold from the 2R chromosome. (**A**) depicts the contigs pertaining to GB-ls-coga4_sup assembly, while (**B**) depicts the contigs particular to GB-ls-coga4_all. Some gene-queries contained repetitive genomic sequences that could be found within more than one contig (blue hits). Since the overall number of blue hits can be a measure of redundant genomic information, the comparative graphics show how the FLO-FLG114 positively influenced the redundancy of GB-ls-coga4_all. The collage was created with BioRender.com (accessed on 23 November 2024).

**Figure 4 insects-16-00002-f004:**
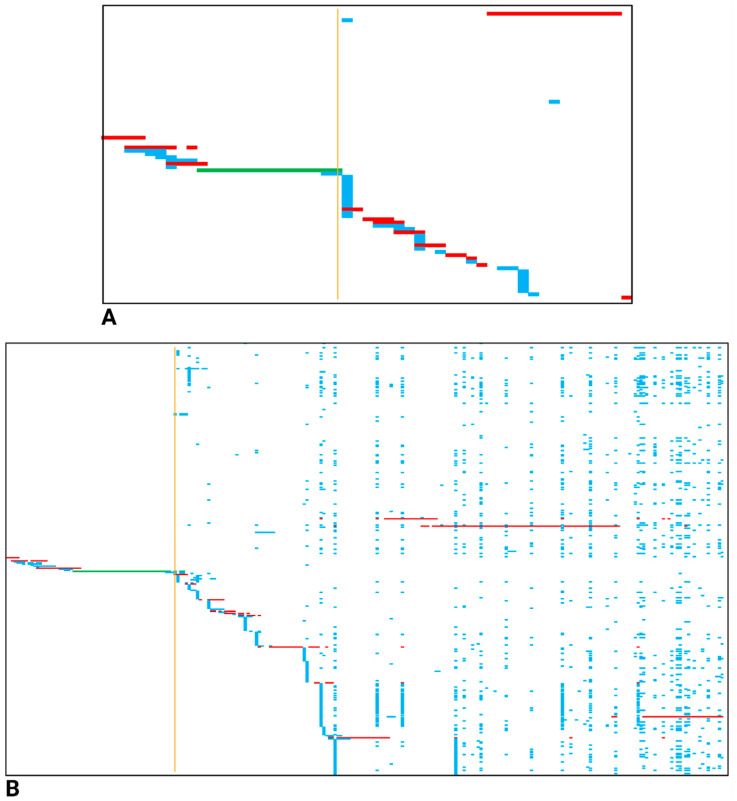
Comparison of results obtained with dScaff when gene-queries (**A**) and ranked-queries (**B**) were employed in order to sort the contigs from GB-ls-coga4_all that match the NW_023496808.1 scaffold. In (**A**), the left or smaller genomic coordinate is 6,819,396, while in (**B**), the corresponding coordinate is 6,813,001. Starting approximately from the 7,353,001 genomic coordinates (vertical orange line), the scaffold seems to be particularly rich in sequence repeats, a feature that is highlighted especially when ranked-queries are used. The collage was created with BioRender.com (accessed on 23 November 2024).

**Figure 5 insects-16-00002-f005:**
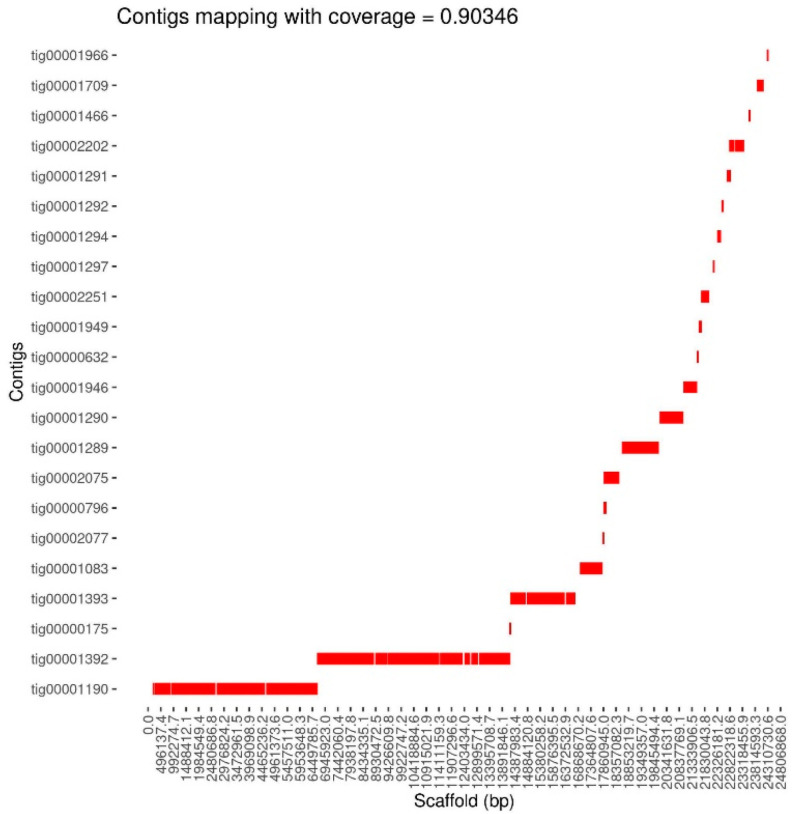
The automatic graphical output for the minimal scaffold corresponding to the 2R chromosome of the GB-ls-coga4 genotype. The contigs name and mapping relative to the reference chromosome are presented on the *Y-* and *X*-axis, respectively. In addition, an approximate estimation of the linear chromosomal coverage of 90.346% is provided on the upper line.

**Figure 6 insects-16-00002-f006:**
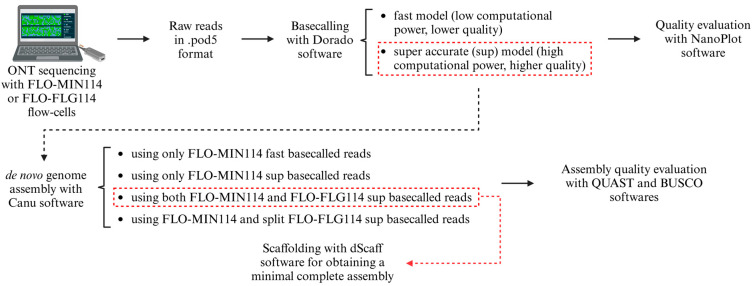
Schematic representations of the pipeline implemented for obtaining optimal assembly quality. The red boxes mark the basecalling model and the assembly with the highest qualities. Starting from the best assembly, we generated a minimal complete assembly using dScaff (indicated by the red arrow). Created with BioRender.com (accessed on 19 December 2024).

**Table 1 insects-16-00002-t001:** Various sequencing parameters characterizing the experiments performed on the GB-ls-coga4 line. Abbreviations for reads and bases refer to their corresponding number (k = kilo, 10^3^, M = Mega, 10^6^, and G = giga, 10^9^). “Number of bases” and “N50 value” columns refer to nucleotide bases (b), and file sizes are shown in gigabytes (GB).

*D. suzukii* Line	Flow-Cell	Input DNA Concentration (fmoles)	Read Count	Number of Bases	N50 Value	File Size
GB-ls-coga4	FLO-MIN114	16.18	4.6 M	16.59 Gb	8.9 kb	198.7 GB
	FLO-FLG114	7.7	116.59 k	328.25 Mb	9.7 kb	3.8 GB
		7.7	76.72 k	242.47 Mb	10.09 kb	2.8 GB
		19.41	224.54 k	347.58 Mb	3.67 kb	4.2 GB
		20	34.94 k	87.12 Mb	7.3 kb	1.0 GB
		20	155.39 k	268.84 Mb	4.04 kb	3.1 GB

**Table 2 insects-16-00002-t002:** Results obtained with QUAST for the four assemblies include the genome fraction, the duplication ratio, the number of contigs, the largest contig, and the N50 value.

Assembly	Genome Fraction (%)	Duplication Ratio	Number of Contigs	Largest Contig (Bases)	N50 (Bases)
GB-ls-coga4_fast	87.969	1.497	2344	6,051,109	554,087
GB-ls-coga4_sup	91.08	1.928	4793	24,357,286	209,854
GB-ls-coga4_all	89.201	1.378	1880	13,127,561	714,986
GB-ls-coga4_polca	90.945	1.924	4793	24,354,269	209,828

**Table 3 insects-16-00002-t003:** The BUSCO results for the five assemblies. “Complete” indicates the overall completeness of the assembly; “Single Copy” refers to the percentage of uniquely identified genes within the assembly, and “Duplicated” indicates the percentage of duplicate genes, reflecting assembly redundancy. “Fragmented” denotes the percentage of partially identified genes and “Missing” represents the percentage of genes apparently absent from the assembly.

Assembly	Complete	Single Copy	Duplicated	Fragmented	Missing
GB-ls-coga4_fast	84.5%	71.6%	12.9%	9.1%	6.4%
GB-ls-coga4_sup	98.6%	66.4%	32.2%	0.8%	0.6%
GB-ls-coga4_all	99.6%	90.1%	9.5%	0.1%	0.3%
GB-ls-coga4_polca	98.8%	67.5%	31.3%	0.6%	0.6%
Dsuz_RU_1.0_all	98.6%	98.3%	0.3%	0.2%	1.2%

**Table 4 insects-16-00002-t004:** The BUSCO results for the minimal assembly of GB-ls-coga4_all, synbolyzedGB-ls-coga4_min, the actual GB-ls-coga4_all assembly, and the variant of Dsuz_RU_1.0 reference assembly, consisting of only the complete chromosomes (Dsuz_RU_1.0_chr). When comparing the minimal and reference assemblies, except for the percentage of duplicated genes, the rest of the assembly features aimed for by BUSCO have similar values. In contrast with the reference chromosomes that are each represented by a single contig, our scaffolding procedure provides a collection of individual contigs, assuring minimal linear coverage of the chromosomes; therefore, the total number is 84, considerably fewer than the complete set of 1880 contigs.

Assembly	Complete	Single Copy	Duplicated	Fragmented	Missing	Contigs
GB-ls-coga4_min	96%	94.7%	1.3%	0.9%	3.1%	84
GB-ls-coga4_all	99.6%	90.1%	9.5%	0.1%	0.3%	1880
Dsuz_RU_1.0_chr	96.5%	96.2%	0.3%	0.6%	2.9%	6

## Data Availability

*D. suzukii* Romanian strain assemblies, namely, GB-ls-coga4-fast and ICDPP-ams-1_sup, are available on GenBank/NCBI as GCA_037074645.1 and GCA_040114545.1, respectively. This Whole Genome Shotgun project has been deposited at DDBJ/ENA/GenBank under accession number JAYGJW000000000. The version described in this paper is version JAYGJW020000000. Developed bioinformatics procedures are open-source and available at: https://github.com/DL-UB/FastqSequenceSplitter (accessed on 1 June 2024), https://github.com/DL-UB/dScaff (accessed on 30 October 2024), and https://github.com/DL-UB/SubSequencesExtractor (accessed on 20 October 2024).

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
