# Peer review of "The Impact of Oxford Nanopore Technologies Based Methodologies on the Genome Sequencing and Assembly of Romanian Strains of *Drosophila suzukii"

_insects, 2024, doi:10.3390/insects16010002_

Round 1

Reviewer 1 Report

Comments and Suggestions for Authors

=== Comments to the authors

The manuscript deals with genomics of an important pest species Drosophila suzukii and technical approaches to genome assembly of Romanian strains. The approach is highly justified for many reasons, from purely technical ones to the fact that this species has many TE elements in its genome (more than other species of this genus). I genuinely enjoyed reading this manuscript and am looking forward to seeing the further analysis with evolutionary conclusions.

The manuscript is well-written, and I particularly liked how diligently the authors retell the experimental and analytical procedures. There are some comment and suggestions (please see below).

It would be helpful for understanding of the story to report the genome size of D. suzukii (and variation between different populations if such information is available) when introducing the species.

L14 “aiming to characterized” : “aiming to characterize”?

L107-108 “banana-yeast-agar-semolina based standard culture medium”: I understand what is meant and totally agree that it is a standard and easy-to-make culture medium, but I really have a feeling that every lab does it different, so it would be great to provide a more detailed recipe for reproducibility.

L438 “dScaff generated “mapped_contigs.csv” file contains tabular data,”: “dScaff generated “mapped_contigs.csv” file containing tabular data,”?

L492-493 “Both BUSCO and dScaff comparisons showed relatively elevated percentages of redundancy within GB-ls-coga4_all, which exhibited the best assembly characteristics.”: Here I’m a bit confused. Do you mean here that redundancy levels were relatively high even in this assembly, which was the best, or that they were elevated relative to some of the other assemblies (this I don’t understand)? Maybe this sentence could be reworded to be understood unambiguously.

Table 4: maybe it would make sense to add GB-ls-coga4_all to this comparison? I had to scroll back to Table 3 to compare. You could highlight the new assembly in bold, for example.

The Discussion section is quite lengthy, and some information is repeated from Results, so it might become hard to read. While I respect the right of the authors to choose whatever style they prefer, I feel that a reader might need a visual help to understand what is going on. A schematic of the most successful pipeline in this section would be a great addition and visual aid.

L703-704 “generated a considerable higher number of alignments”: “generated a considerable higher number of alignments”?

L714-715 “our tactic relying on evaluating BLAST alignments to not be suited”: “our tactic relying on evaluating BLAST alignments is not suited”?

L717 “In addition to facilitate”: “In addition to facilitating”?

Author Response

The responses for reviewer 1 are in the attached document.

Reviewer 2 Report

Comments and Suggestions for Authors

The manuscript by Ratiu, Ionascu and Constantin entitled “The impact of Oxford Nanopore Technologies based methodologies on the genome sequencing and assembly of Romanian strains of Drosophila suzukii” presents the genomic sequence of two lines of D. suzukii obtained from two near locations in Bucharest. For each strain, two different ONT sequencing libraries were obtained, the first one was loaded in a MinION Flow-Cell R10, and the second to five different Flongle Flow-Cell R10. Different assembly strategies were performed, and they are tested for completeness and accuracy relative to the reference genome available on NCBI.

Comments to the authors:

I think that the D. suzukii husbandry performed do not let assume that the lines had meet the criterion of near isogenic lines. 

The text says:

Prior to genomic sequencing, both D. suzukii lines were kept in laboratory settings over a year, thus we expected from them to meet the criterion of near isogenic lines with a relatively homogenous genomic background.” 

I have doubts about the isogenicity characteristic of these lines. When one wants to obtain an isogenic line, one usually starts with a single fertilized female (4 haploid genomes) and in each generation (for 13-15 generations), controlled crosses between siblings are performed. Even then, there is some residual heterozygosity. In the D. suzukii husbandry explained, the line ICDPP started with two females (and two males) so at least 8 haploid genomes, and in the case of line GB “few” males and females I suspect there would be more. If the mates are not controlled, each generation will have more variability due to recombination.

As explained in lines 181-184, for each line, the first genome assemblies were using FLO-MIN114 data handled with both fast and sup basecalling models. My question is why the submissions to NCBI for each line correspond to a different basecalling model?

Figure 1. For completeness, the two locations should be identified with a letter and an explanation in the figure legend. 

Table 1. I wonder why the results of FLO-MIN114 replicates using the same input DNA concentration are so different. For example, starting with 20 fmoles the second sample yields three times the number of bases as the first. OK, eight pages after in Discussion section this observation is also noted but no explanation is given.

L.540-541 “From the start, our aim was to sequence, assemble and annotate the two genotypes,” I think that here “genotypes” must be changed to “genomes”. When we talk about the genotype of a diploid organism, we are referring to the variants present on its two homologous chromosomes. Although the authors stated earlier that they assumed these lines are nearly isogenic, there must still be a lot of residual heterozygosity present and nowhere is there any reference to variants within each strain.

Minor comments and typos:

L.13 Change “The” to “This”

L.15 Change “the” to “this”

L. 331 Change “from either fast of” to “from either fast or”

L. 715. I had problems understanding the text. Better change “BLAST alignments to not be suited for” to “BLAST alignments is not suitable for”

References. Please check throughout the section that species names are written in italics.

Supplementary Table 2. Please indicate (as in Tables S4 and S5) which columns are log2 transformed.

Figure legend of Supplementary Figure 2. I think that “tig00001376 contig” should be changed to “00001376 contig”

Author Response

The responses for reviewer 2 are in the attached document.
